# Investigation of the Structure and Allergic Potential of Whey Protein by Both Heating Sterilization and Simulation with Molecular Dynamics

**DOI:** 10.3390/foods11244050

**Published:** 2022-12-14

**Authors:** Zhao Zhang, Ruida Ma, Yunpeng Xu, Lei Chi, Yue Li, Guangqing Mu, Xuemei Zhu

**Affiliations:** 1School of Food Science and Technology, Dalian Polytechnic University, Dalian 116034, China; 2Dalian Women and Children Medical Center, Dalian 116012, China

**Keywords:** whey protein isolate, heat sterilization, structure and allergy, molecular dynamics simulation

## Abstract

As the main allergens in milk, whey proteins are heat-sensitive proteins and are widespread in dairy products and items in which milk proteins are involved as food additives. The present work sought to investigate the effect of heating sterilization on the allergenicity of α-lactalbumin (α-LA) and β-lactoglobulin (β-LG), the main composite and allergen in whey protein isolate (WPI), by combining molecular dynamics with experimental techniques for detecting the spatial structure and IgE binding capacity. The structure of WPI was basically destroyed at heat sterilization conditions of 95 °C for 5 min and 65 °C for 30 min by SDS-PAGE analysis and spectroscopic analysis. In addition, α-lactalbumin (α-LA) may be more sensitive to temperature, resulting in exposure to allergic epitopes and increasing the allergic potential, while the binding capacity of β-lactoglobulin (β-LG) to IgE was reduced under 65 °C for 30 min. By the radius of gyration (Rg) and root-mean-square deviation (RMSD) plots calculated in molecular dynamics simulations, α-LA was less structurally stable at 368 K, while β-LG remained stable at higher temperatures, indicating that α-LA was more thermally sensitive. In addition, we observed that the regions significantly affected by temperatures were associated with the capacity of allergic epitopes (α-LA 80–101 and β-LG 82–93, 105–121) to bind IgE through root-mean-standard fluctuation (RMSF) plots, which may influence the two major allergens. We inferred that these regions are susceptible to structural changes after sterilization, thus affecting the allergenicity of allergens.

## 1. Introduction

Milk is regarded as nature’s ideal food because it contains almost all the nutrients essential for humans, but it is also one of the most common allergens to cause allergic reactions [1,2]. To date, studies have shown that the incidence of allergic reactions in infants and young children due to milk intake is increasing, and in some cases, the intake of milk and its products is accompanied by severe clinical symptoms, such as itchy skin and dyspnea, which can be life-threatening [3]. In addition, milk protein allergies can lead to the destruction of intestinal villi, which leads to a lack of lactase and lactose intolerance. The complications of milk protein allergies may also include nutritional and growth impairment, anemia, and insufficient bone calcification [4]. Whey protein isolate (WPI), a class of protein that accounts for about 20% of total cow’s milk protein, contains two major allergens of α-lactalbumin (α-LA) and β-lactoglobulin (β-LG). β-LG is more allergenic than α-LA, and about 80% of milk allergy patients are allergic to β-LG due to it not existing in human milk and maintaining a relatively stable structure in the strongly acidic environment of gastric acid [5,6].

The food processing operations that are used to maintain food safety and quality are varied, involving many processing conditions, such as temperature, pressure, and pH value [7]. These parameters, including temperature and time, have a significant effect on the structure of whey protein. Heat treatment directly affects the allergenicity of whey proteins by changing the epitopes that whey protein is involved with in an immune reaction. The effects of heat treatment on the epitopes of whey protein include burial, exposure, and fragmentation, which may significantly reduce the allergenicity of allergens [8]. Glycation treatment buries some of the immunodominant IgE allergenic epitopes by inducing β-sheet and β-turn changes, resulting in reduced ovalbumin allergenic reactivity [9]. However, the structural changes of allergens caused by heat treatment may also form new conformational epitopes [10]. Heating causes persistent structural changes for peanut allergen Ara h 2 and leads to the formation of Ara h 2 oligomers in solution. CD spectra revealed that the heating of recombinant, as well as natural, Ara h 2 led to conformational changes, and the heated protein exhibited a significantly higher immunogenic potential [11]. In general, conformational epitopes are more easily influenced by heating than linear epitopes, though changes in linear epitopes can also be induced under extreme heat treatment conditions [12]. Moreover, some physical and chemical properties of whey protein amino acid residues are altered by heat treatment [13], which may lead to serious effects, such as in the interaction with other components in product and whey protein gastrointestinal digestion, thus regulating allergies [14].

At present, existing experimental methods can characterize the spatial structure and kinetic characteristics of proteins by, for example, X-ray diffraction, atomic force microscope (AFM), circular dichroism, and fluorescence analysis, but it is impossible to intuitively display the changes in the spatial structure of proteins. The emergence of molecular dynamics (MD) can not only guide the experiment under the premise of cost saving but also visualize structural changes in allergic proteins under different conditions. MD simulations, as a standard tool for studying biomolecules, can effectively help us understand biochemical processes and fully express the macroscopic properties of a system changing with time [15]. Garrido-Arandia et al. [16] explored three major allergens in Alternaria mold, birch pollen, and peach fruit by MD simulations and found that the flexibility of the allergy epitope residue was lower than or similar to that of the protein as a whole in spite of spanning part of the loop regions. MD simulations revealed that the secondary and tertiary structures of ovomucoids were significantly altered by heat treatment when the temperature was increased from 25 to 100 °C with the exposure of linear epitopes (K159–S174 and T179–C186) [17]. In addition, the conformation of the epitope position also changed. Similarly, the exposure of linear epitopes and the change in the conformational epitopes of gliadins were highlighted after the protein was heated from 25 °C to 100 °C by MD simulation [18].

Pasteurization is a common sterilization method for dairy production, typically including high-temperature, short-time (80 °C for 15 s) and low-temperature, long-time (65 °C for 30 min) sterilization, and sterilization at 95 °C for 5 min is a common process for sterilizing milk in yogurt production in China. It is critical to understand the variation in milk protein allergenicity in the actual production of dairy products. Therefore, in this study, the effects of different heating sterilizations on the structure of major allergens in whey proteins were investigated by using spectroscopy, scanning electron microscopy (SEM), and reductive SDS-PAGE in combination with MD simulations. We used competitive enzyme-linked immunosorbent assay (ELISA) to assess the allergic potential of allergens after treatment with different methods, and the effect of different temperatures on the major epitopes of allergens was analyzed by MD simulation. This will provide a theoretical basis for the production of low-allergic dairy products under common sterilization methods.

## 2. Materials and Methods

### 2.1. Materials

The whey protein isolate (WPI, 80% protein content, Yuanye Co., Shanghai, China) was dissolved in deionized water (protein concentration was 6.6 mg/mL) and sterilized by different methods (65 °C for 30 min, 80 °C for 15 s, and 95 °C for 5 min).

The serum pool was prepared by mixing serum from 15 donors with cow’s milk allergy in equal volumes (Table 1), which were collected from Dalian Municipal Women and Children’s Medical Center Group (Hope Square Branch, Dalian, China).

### 2.2. Ultraviolet–Visible (UV) Spectrum Analysis

The expansion of WPI after sterilization was detected by UV spectrophotometer (UV-2600, Shimadzu Co., Kyoto, Japan). The wavelength scanning range was 190–900 nm. The control group was untreated WPI. 

### 2.3. Fluorescence Spectrum Analysis

The structural changes in WPI after sterilization were detected by fluorescence spectrophotometer (RF-6000, Shimadzu Co., Kyoto, Japan). The wavelength scanning range was 250–350 nm. The slit was 5 nm, the acquisition interval was 0.1 nm, and the scanning speed was slow [19]. The control group was untreated WPI. 

### 2.4. Circular Dichroic (CD) Spectral Scanning

The secondary structure of WPI after sterilization was detected by CD spectrometer (J-1500-150, Jasco Co., Tokyo, Japan). The wavelength scanning range was 190–260 nm. The sample concentration of whey protein solution to be measured was 0.25. The acquisition interval was 1 nm [20]. The control group was untreated WPI.

### 2.5. Scanning Electron Microscope (SEM) Analysis

The microstructure of WPI after sterilization was observed by SEM (JSM-6460, JEOL, Tokyo, Japan). The magnification was ×30,000, ×2000, and ×5000 [21].

### 2.6. Reductive SDS-PAGE Analysis

The molecular weight changes in WPI after sterilization were analyzed by SDS-PAGE. The manufacture of the discontinuous gel system consists of acrylamide stacking gel (5%, w/v) and acrylamide separating gel (12%, w/v). Each sample was mixed with the buffer solution (in the presence of 5% β-mercaptoethanol) at 1:1 and heated for about 5 min at 100 °C, then loaded into different slots. Each slot was 10 μL. Following sample addition, constant voltage (concentrated gel 90 V, separation gel 150 V) electrophoresis was carried out. After electrophoresis, the gel was stained with Coomassie Brilliant Blue R250 for 3–4 h, then decolorized in decolorizing solution (C_2_H_5_OH 500 mL, CH_3_COOH 90 mL, H_2_O 410 mL) until the background was clear. Scanning and imaging were performed by the C300 chemiluminescence imaging system (Azure Biosystems, Dublin, CA, USA) [22].

### 2.7. Determination of Binding Capacity with IgE

The capacity of both α-LA and β-LG in WPI to bind specific IgE was assessed by indirect competitive ELISA according to the method of Liu et al. [23] with slight modifications. α-Lactalbumin (L6010, purity 85%, Merck KGaA, Darmstadt, Germany) and β-lactoglobulin (L3908, purity 90%, Merck KGaA, Darmstadt, Germany) were dissolved in carbonate buffer solution (CBS, 0.05 mol/L, pH 9.6) at a concentration of 50 μg/mL and coated in ELISA plates, respectively, and serum diluted 20 times. Goat anti-human IgE antibody, biotin conjugate (Thermo Fisher Scientific, Shanghai, China) was dissolved in phosphate-buffered saline (PBS, 0.1 mol/L, pH 7.4) at a ratio of 1:2000. HRP-streptavidin (Solarbio, Beijing, China) was dissolved in PBS at a ratio of 1:2000.

OD values at 450 nm and 630 nm were measured on a microplate reader (Thermo Scientific, Massachusetts, USA). The inhibition rate was calculated according to the following formula:ΔOD = OD_450_ − OD_630_(1)
Inhibition rate (%) = (ΔOD_0_ − ΔOD)/ΔOD_0_ × 100(2)

ΔOD: The result of ELISA was the difference in the optical density of samples.

ΔOD_0_: The difference in optical density of noncompetitive.

### 2.8. MD Simulations

The tertiary structures of α-LA and β-LG were obtained from the Research Collaboration for Structural Bioinformatics Protein Data Bank (RCSB PDB, https://www.rcsb.org/ (accessed on 11 May 2022)), and their PDB IDs are 1F6S and 3NPO, respectively.

The MD simulation was performed by GROMACS 2022.1(GROMACS 2022.1, GROMACS development team, Groningen, Netherlands) and used the charmm 27 force field. The PDB files were validated to confirm that all necessary atoms were present. Topology files were produced through the module pdb2gmx; the water model SPC216 was used to solvent the protein, and the electrically neutral system was used for MD simulations by adding ions according to the charges present on the protein. The structure was then relaxed through an energy minimization (EM) process (at a maximum force of 1000.0 KJ/mol/nm by using 50,000 steps).

According to the sterilization temperature, the temperatures were set to a maximum of 298 K (25 °C), 338 K (65 °C), 353 K (80 °C), and 368 K (95 °C), respectively, and the temperature of the system was stabilized by NVT equilibrium of 100 ps, and then the NPT equilibrium of 100 ps was carried out to stabilize the pressure of the system. After the two equilibration phases were completed, MD was run for data collection. Finally, the stability of the protein was observed by 1 ns MD simulation. The gyrate, root-mean-square deviation (RMSD), and root-mean-standard fluctuation (RMSF) were calculated and then plotted in 2D using origin 2019 software (Origin 2019, OriginLab Co., Northampton, MA, USA). In addition, the structural differences between the two proteins at different temperatures were shown by PyMOL software (PyMOL 2.5.2, Schrödinger, New York, NY, USA) with 3D diagram. In addition, the changes in the major allergic epitopes of the two proteins were analyzed according to the results of MD.

### 2.9. Statistical Analysis

All tests were repeated six times. Statistical analysis of the data was performed using SPSS 22.0 software (SPSS 22.0, International Business Machines Co., New York, NY, USA). The significance of the data was assessed by Duncan’s multiple range test with a significance level of *p* < 0.05.

## 3. Results

### 3.1. SDS-PAGE Analysis

The molecular weight distribution of WPI after sterilization under different conditions is shown in Figure 1. The bands of α-LA became darker in the sterilized samples, and contrarily, the bands of β-LG became lighter. This suggests that the structure of α-LA was altered after heat treatment, resulting in changes in protein content upon reduction.

Following heat treatment to lighten β-LG bands, the bands in the loading hole became clear, and the color gradually deepened with an increase in sterilization temperature, indicating that heating made WPI form a polymer and it, therefore, could not enter the stacking gel [24]. Native β-LG exists as a dimer, and its thermal denaturation temperature is higher than that of α-LA. After heating, β-LG is depolymerized into monomers and hydrolyzed into peptides, exposing the reactive -SH inside the molecule to form hydrophobic interactions to promote the aggregation reactions [25,26]. In addition, α-LA with thermal denaturation also exposes the intramolecular disulfide bonds and polymerizes with the reactive -SH of β-LG to form aggregates [26]. Dalgleish et al. [27] confirmed that β-LG plays a key role in polymerization.

### 3.2. Spectroscopic Analysis of Whey Protein

#### 3.2.1. UV Spectrum Analysis

The UV-Vis absorption spectra of WPI after treatment with different heating sterilization are shown in Figure 2A. Compared with the original sample (303 nm), the UV absorption peaks of the sterilized sample were redshifted to 306 nm (80 °C for 15 s), 310 nm (65 °C for 30 min), and 319 nm (95 °C for 5 min), respectively. The occurrence of redshift may be due to the effect of sterilization on the structure of WPI, in which hydrophobic amino acids were buried, and the hydrophobic interactions between hydrophobic groups were weakened, resulting in some changes in molecular structure [28]. We observed that the UV absorption peak of the sterilized WPI was higher than that of the untreated sample, indicating that heating could expand the structure of WPI.

#### 3.2.2. Fluorescence Spectrum Analysis

As shown in Figure 2B, the maximum fluorescence intensity of unsterilized WPI was 140.69 (367 nm). Compared with the untreated samples, the three sterilized samples, 80 °C for 15 s was 100.39 (368 nm), 65 °C for 30 min was 89.24 (369 nm), and 95 °C for 5 min was 69.20 (371 nm), showed a slight redshift and changes in fluorescence intensity. A possible reason for the redshift in the maximum fluorescence intensity peak is that heating sterilization buries the side chains of these amino acid residues and increases the polarity of the system [29,30]. The spontaneous fluorescence of aromatic amino acids encapsulated by nonpolar groups in the whey protein molecule is present inside the molecule. Moreover, the fluorescence intensity of WPI was significantly reduced by sterilization, in which the effect of sterilization at 95 °C for 5 min was the most obvious, indicating that heating folded the tertiary structure, burying hydrophobic amino acids and tryptophan residues inside the protein, ultimately leading to an increase in the polar microenvironment of WPI.

#### 3.2.3. CD Spectrum

The CD spectrum of untreated WPI (shown in Figure 2C) exhibited two positive peaks at 190 and 207 nm and two negative peaks at 195 and 223 nm. These phenomena indicate that the α-helix and β-sheets are rich in the secondary structure of whey protein. As shown in Figure 2C, sterilization leads to a decrease in the band intensities of CD spectra without a significant shift in the peaks, and this phenomenon becomes more obvious with an increase in sterilization temperature and time (65 °C for 30 min, 80 °C for 15 s, and 95 °C for 5 min). This phenomenon suggests that sterilization has a dramatic effect on the α-helix and β-sheets in whey proteins. This may be because many hydrogen bonds inside the α-helix can react with polar groups, thus affecting the percentage of α-helix [31].

From the results of the UV, fluorescence, and CD spectrum, we conclude that the spatial structure of WPI is less affected by sterilization at 80 °C for 15 s and the secondary structure is less affected by sterilization at 65 °C for 30 min. Interestingly, the degree of structural change in WPI not only intensified exclusively with increased temperature but was also positively related to the time of sterilization. However, higher temperatures changed the protein structure, in which the allergic epitopes were possibly more exposed or buried, which may affect the allergenic potential of WPI.

### 3.3. Microstructure Analysis by SEM

The spherical and planar structures of the unsterilized WPI were intact, and the surfaces were smooth (Figure 3). In contrast, the spherical structures of the samples sterilized at 65 °C for 30 min and 95 °C for 15 s showed holes and depressions on the surface, and the planar structures were destroyed. Furthermore, the structure of the samples sterilized at 95 °C for 15 s was more severely damaged than in other conditions. However, the spherical structure of the sample sterilized at 80 °C for 15 s was minimally changed, and there were only a few holes in the planar structure, but it was not completely destroyed. These were consistent with the results of the optical analysis. 

### 3.4. Analysis of Binding Capacity to IgE

The capacity of the WPI sample to inhibit the binding of standard proteins (α-LA and β-LG) to IgE reflects the allergic potential of the sample (Figure 4). For sterilized samples, the inhibition rate of allergen α-LA was significantly higher than that of untreated WPI (*p* < 0.05), indicating that the binding capacity of α-LA to IgE was enhanced after sterilization. The most significant increase in the inhibition rate reached 83.40% under sterilization at 95 °C for 5 min (*p* < 0.05), which was 71.68% higher than that without sterilization. Toda et al. [32] studied the effect of heat treatment on the allergenicity of α-LA in the temperature range of 50–121 °C. The results showed that the maximum allergenicity of α-LA was reached at 90 °C, which is similar to the results of the present experiment. In addition, there was no significant difference in the inhibition rate between the two samples sterilized at 65 °C for 30 min and 80 °C for 15 s (*p* > 0.05), indicating that prolonged heat treatment is also able to lead to more exposure of allergic epitopes, resulting in an increase in allergenicity [33]. Therefore, for this phenomenon, a longer treatment time can make up for the disadvantage of temperature.

The effects of sterilization at 65 °C for 30 min and 80 °C for 15 s on the inhibition rate of β-LG were not significantly different compared with unsterilized WPI (*p* > 0.05); the inhibition rate decreased significantly at 95 °C for 5 min (*p* < 0.05).

Toda et al. [32] confirmed that when the heat treatment temperature is lower than 80 °C, there is no significant difference in the allergenicity of β-LG, but the allergenicity of β-LG decreases significantly when the heat treatment temperature is above 90 °C. Furthermore, our results show that temperature is an important factor affecting the allergenicity of β-LG, including the denaturation of β-LG, destruction of the allergenic epitope, and a decrease in the binding capacity to IgE at higher temperatures, which results in the same conclusion as the study by Toda et al. [32]. Notably, Kleber et al. [34] showed that there is a slight increase in the allergenicity of β-LG before the β-LG denaturation temperature is reached. This is caused by the unfolding of the conformational structure of β-LG during heat treatment, which exposes the allergic epitopes inside the molecule.

### 3.5. MD Simulations of Major Allergens

#### 3.5.1. α-LA

The rather small amplitude of the gyration radius (Rg, Figure 5A) indicates that the protein size remains stable during the simulation time [16,35], during which the fluctuation of Rg for α-LA at 368 K was slightly more severe with a standard deviation of 0.01 nm. In addition, the larger the Rg, the greater the expansion of the system. During the simulation, there was almost no difference in the average protein expansion level at the four temperatures. However, the expansion degree of α-LA was greatest at 368 K, and the Rg showed a maximum at 720 ps, which was 1.40 nm. Figure 5B shows the trend of the RMSD of the protein with the simulation time. We observed that before 0.1 ns, the RMSDs of the three groups were in relative equilibrium except at 368 K. Furthermore, at the later stage of the simulation, the RMSD values at 368 K were higher, indicating that the conformation of the protein was significantly greater and the stability was poor [36].

The RMSF plots calculated for the residue of α-LA in the MD simulations for all temperatures are shown in Figure 6A. At high temperatures, the regions covered by most of the residues displayed small fluctuations compared with those at room temperature (298 K), except for the region of 80–101. Using PyMOL (Figure 6B), we showed that the region of 80–101 was mainly composed of α-helix and loops and confirmed that the region changed. The loops are a flexible structure, which is easily changed by the environment; there are many hydrogen bonds within the α-helix, and they are involved in some chemical reactions, which may lead to a change in the α-helix structure [37]. In conclusion, these changes were consistent with the results of the RMSF plots. Notably, the region of 80–101 contained some allergic epitopes [38,39], so it could be speculated that high temperature may affect the conformation of allergic epitopes in this region, thus affecting the sensitization allergy of α-LA.

#### 3.5.2. β-LG

During the MD simulations, the Rg plots (Figure 7A) of β-LG at four temperatures were all stable, as were the RMSD plots (Figure 7B), indicating that there was less change in the structure of the protein. Furthermore, compared with the plots of α-LA, it could also be assumed that β-LG was less sensitive to heat than α-LA in a shorter period of time.

The RMSF plots of β-LG amino acid residues are shown in (Figure 8A). Compared with at room temperature (298 K), β-LG underwent significant changes in two regions at the other three higher temperatures, region 82-93, and region 105–121, respectively. Using PyMOL (Figure 8B), it was shown that the regions 82–93 and 105–121 were both composed of a loop connecting two β-sheets. The regions with greater variations in RMSF values were concentrated in the residues on loops rather than on the β-sheet due to the fact that the β-sheet is a rigid secondary structure. Notably, the peptide segment between positions 82 and 90 in the region 82–93 is part of the major epitope that binds to IgE [40] as well as the region 105–121 [41].

## 4. Conclusions

In this study, we confirmed that three heating sterilizations in common dairy products affected both the structure and allergenic potential of WPI. There was little effect on the spatial structure and microstructure of WPI by sterilization at 80 °C for 15 s, and the other two sterilizations essentially destroyed the structure of WPI. In addition, compared with β-LG, α-LA was more sensitive to temperature, and its structure changed severely, affecting the allergy of α-LA. We showed that both the heat treatment conditions and the types of protein are all factors that affect the allergenic potential of WPI. The results of the molecular dynamics showed that the structural stability of α-LA is poor at 368 K, while β-LG remained stable at higher temperatures, indicating that α-LA is more thermally sensitive. The RMSF plots found that changed regions were associated with IgE binding capacity. We infer that these regions are susceptible to structural changes after sterilization, thus affecting the allergenicity of allergens. This provides a theoretical basis for the production of low-allergic dairy products using common sterilization methods.

## Figures and Tables

**Figure 1 foods-11-04050-f001:**
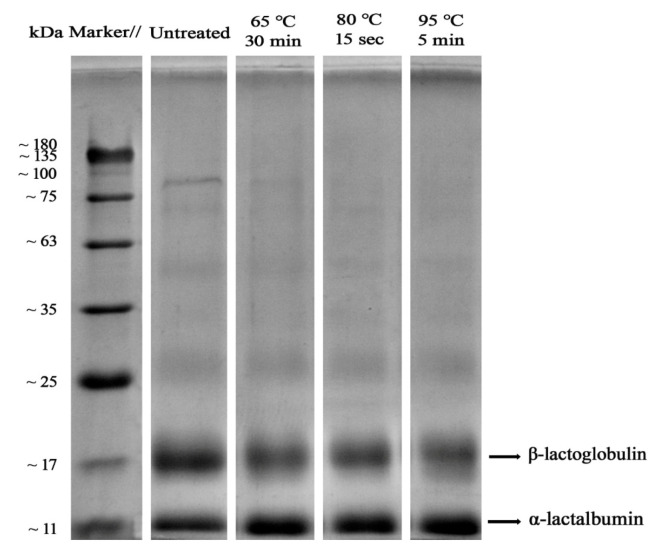
SDS-PAGE analysis of WPI with different sterilizations.

**Figure 2 foods-11-04050-f002:**
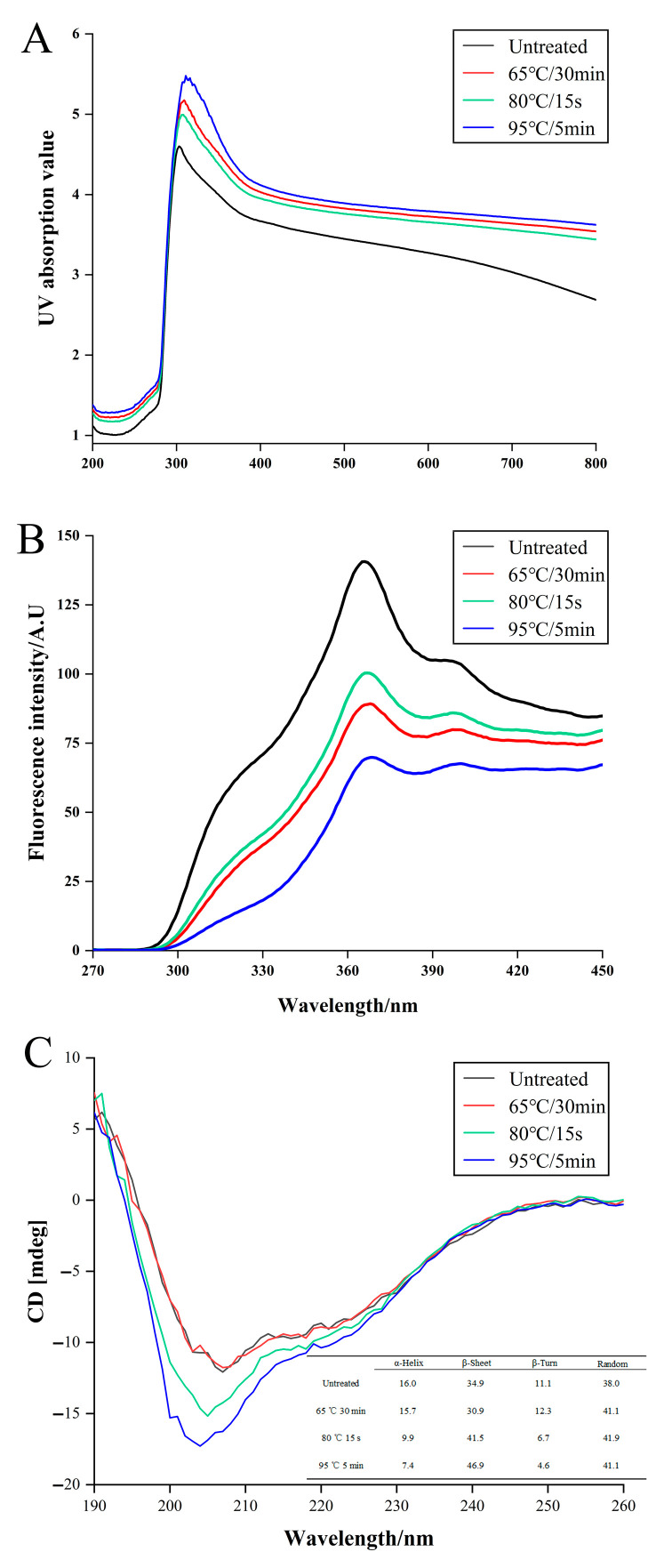
Structural changes in WPI after different sterilizations by spectroscopic analysis. The analysis of spatial structure by (**A**) UV spectrum and (**B**) fluorescence spectra and secondary structure by (**C**) CD spectra.

**Figure 3 foods-11-04050-f003:**
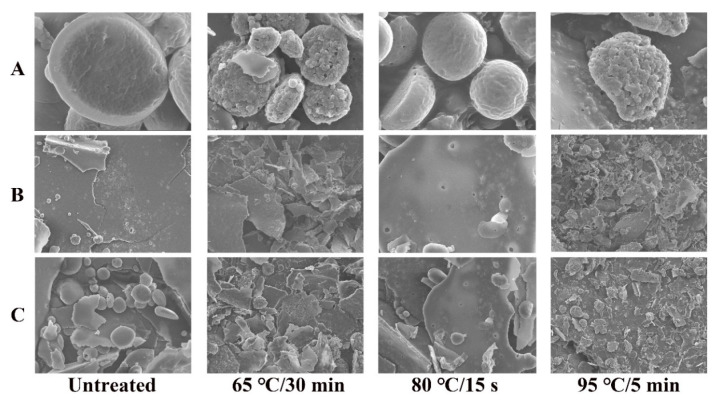
Effects of different sterilizations on the microstructure of WPI by SEM. The magnification is ×30,000 (**A**), ×2000 (**B**), and ×5000 (**C**), respectively.

**Figure 4 foods-11-04050-f004:**
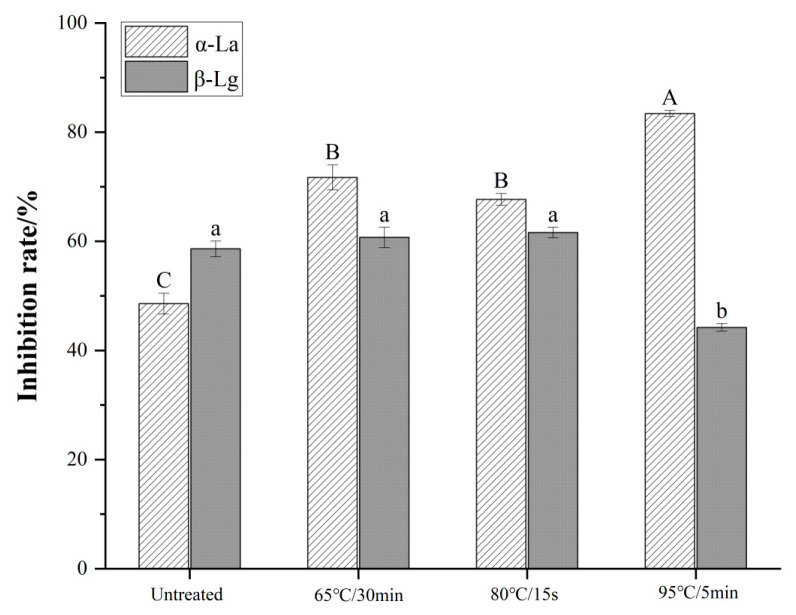
Effects of different sterilizations on the binding capacity of major allergens and specific IgE in WPI. Different uppercase letters indicate significant differences in α-LA after different treatments (*p* < 0.05). Different lowercase letters indicate significant differences in β-LG after different treatments (*p* < 0.05).

**Figure 5 foods-11-04050-f005:**
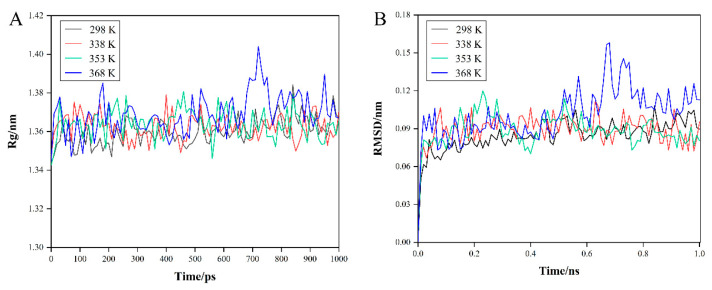
MD simulates the differences of α-LA in different temperature environments, and (**A**) Rg plot and (**B**) RMSD plot.

**Figure 6 foods-11-04050-f006:**
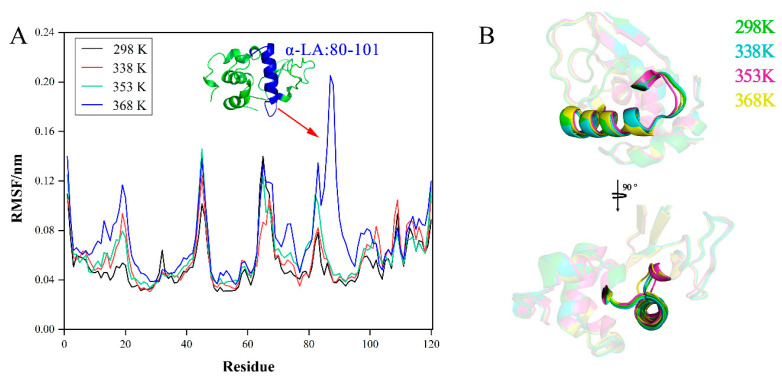
MD simulates the stability of allergic epitopes of α-LA in different temperature environments: (**A**) RMSF plot of protein, and (**B**) conformational differences in region 80–101 in different environments. The arrow indicates the amino acid sequence and protein structure of this position.

**Figure 7 foods-11-04050-f007:**
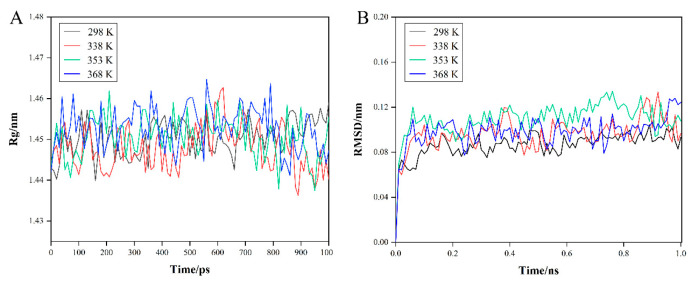
MD simulates the differences of β-LG in different temperature environments, and (**A**) Rg plot and (**B**) RMSD plot.

**Figure 8 foods-11-04050-f008:**
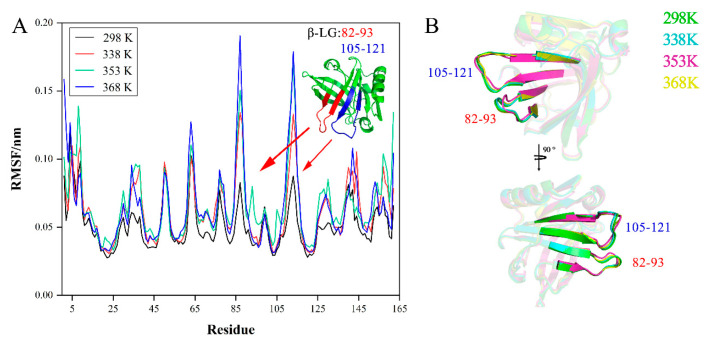
MD simulates the stability of allergic epitopes of β-LG in different temperature environments: (**A**) RMSF plot of protein, and (**B**) conformational differences in regions 82–93 and 105–121 in different environments. The arrow indicates the amino acid sequence and protein structure of this position.

**Table 1 foods-11-04050-t001:** The information of donors with cow’s milk allergy.

Number	Gender	Age	Allergy Clinical Symptoms	Specific IgE Levels in Serum (IU/mL)
1	Female	8 months	Atopic dermatitis	4.646
2	Male	4 years	Asthmatic bronchitis	11.459
3	Female	4 years	None	4.078
4	Male	3 years	Childhood asthma	4.121
5	Male	1 year	Atopic dermatitis	5.203
6	Male	3 years	Eczematous Dermatitis	15.803
7	Male	6 years	Nausea	3.504
8	Male	2 years	Abnormal weight gain	6.184
9	Male	3 years	Physical examination	7.944
10	Male	7 years	Stomachache	5.958
11	Female	2 years	Rash	6.098
12	Male	6 years	Acute rhinitis	7.008
13	Male	2 years	Nosebleeds	17.075
14	Male	4 years	Cough	3.769
15	Female	3 years	Acute bronchitis	4.683

## Data Availability

The data presented in this study are available on request from the corresponding author.

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
