# Peer review of "Investigation of the Structure and Allergic Potential of Whey Protein by Both Heating Sterilization and Simulation with Molecular Dynamics"

_foods, 2022, doi:10.3390/foods11244050_

Round 1

Reviewer 1 Report

I reviewed the manuscript entitled, Investigation of structure and allergic potential of whey protein by both heating sterilization and simulation with molecular dynamics. The concept of this research is novel and contributes to the field. In my opinion, this version of the manuscript can be accepted for publication after addressing the comments below. Many methodologies are without appropriate reference. Authors must improve the discussion in some of the sections

The manuscript must be formatted according to the Foods format. Figures quality must be improved

Line 17: whey proteins (WPI)…. Whey protein isolate?

Lines 51-53: please provide citation

Line 63: whey proteins (WPI)…. Whey protein isolate?

Why did the authors consider only WPI? What about WPC?

Please provide the ref for 2.3 Fluorescence spectrum analysis

please provide the ref for Circular-Dichroic (CD) spectral scanning

Provide the details of SDS-PAGE methodology

3.1 SDS-PAGE analysis: Discussion must be improved

3.2.2 Fluorescence spectrum analysis: discussion must be improved

The quality of the Figure 5 should be improved

Conclusions should be revised according to the findings. As such, it is too wordy

References are not according to the journal format

Overall, the manuscript must be formatted according to the foods format. 

Reviewer 2 Report

The manuscript entitled "Investigation of structure and allergic potential of whey protein by both heating sterilization and simulation with molecular dynamics" is a novel subject showing the effect of treatments on allergenicity and structural properties of whey protein. The purpose of this research is well described. However, below mentioned comments should be addressed:

-The abstract is full of grammatical mistakes and too long (more than 350 words). The abstract should be as concise, representative, and attractive as possible.

-Lines51-53: Please add a reference

-Lines 53-57: please bring more studies revealing this statement 

-Lines 57-59: the authors have mentioned that "The proteins of mammalian milk are homologous, with similar structure, function and biological characteristics". First of all you should know that the structural, functional and biological properties of milk proteins are NOT similar. Secondly, to state such a scientific hypothesis needs a strong and valid scientific base to refer. I think it is better to remove this statement.

-Line 63: WPI is the abbreviation of the whey protein isolate 

-Line 67-68: should be re-written.

-Line 72: "it is almost impossible 72 to avoid heat treatment." ????Why? delete this.

-The introduction is too long and it is not appropriately presented.

-Line 138: you have used the same method of heating with three different time and temperature parameters. Not three methods.

-Line 142: mention the name of the city and the country

-What is Table 1 for? ??where did you use the serum taken from donors????

-Statistical analysis should be described as the last part of material methods right before the results and discussion

-Line 233-236: it is not clear ? why the authors are discussing the stacking gel? you should discuss the bands in the resolving gel. and this section needs more references of other studies to compare.

-The conclusion is too long. I think this manuscript has been copied from a thesis. It would be better if you make it a concise and very well representative single paragraph. not in 2 pages

Round 2

Reviewer 2 Report

The authors have done so much efforts to improve the manuscript. Well done.

I think this is appropriate to accept now.